# Promoting the implementation of clinical decision support systems in primary care: A qualitative exploration of implementing a Fractional exhaled Nitric Oxide (FeNO)-guided decision support system in asthma consultations

Kate Morton[1]*, Marta Santillo[2], Michelle Helena Van Velthoven[3], Lucy Yardley[4,5], Mike Thomas[6], Kay Wang[2], Ben Ainsworth[4], Sarah Tonkin-Crine[2,7]

1 Health Sciences, University of York, York, United Kingdom, 2 Nuffield Department of Primary Care Health Sciences, University of Oxford, Radcliffe Observatory Quarter, Oxford, United Kingdom, 3 Dutches Consulting Ltd, Cambridge, United Kingdom, 4 Faculty of Environmental and Life Sciences, Centre for Clinical and Community Applications of Health Psychology, School of Psychology, University of Southampton, Southampton, United Kingdom, 5 School of Psychological Science, University of Bristol, Bristol, United Kingdom, 6 Primary Care, Population Sciences and Medical Education (PPM), University of Southampton, Southampton, United Kingdom, 7 NIHR Health Protection Research Unit in Healthcare Associated Infections and Antimicrobial Resistance, University of Oxford, Oxford, United Kingdom

* kate.morton@york.ac.uk

## Abstract

### Background

Clinical decision support systems (CDSSs) can promote adherence to clinical guidelines and improve patient outcomes. Exploring implementation determinants during the development of CDSSs enables intervention optimisation to promote acceptability, perceived appropriateness and fidelity during subsequent implementation. This study sought to explore how clinicians perceive the use of a CDSS which makes recommendations for asthma management based on factors including Fractional exhaled Nitric Oxide testing, and how CDSSs can be designed to promote their implementation.

### Methods

Twenty-three interviews were conducted with clinicians to explore perceptions about the CDSS. Participants included asthma nurses, pharmacists, General Practitioners and respiratory nurse specialists involved in conducting asthma reviews in primary care. Interviews were transcribed verbatim and analysed using reflexive thematic analysis.

### Results

Three themes were developed: Appreciating the recommendations of the CDSS, whilst wanting to retain control; Doubt about appropriateness of CDSS recommendations,

**Data Availability Statement:** Raw data in the form of verbatim interview transcripts cannot be shared publicly because of risk of participant identification. Coding manuals, including example quotes, can be made available. For requests to access data, please contact data.protection@admin.ox.ac.uk.

**Funding:** This study summarises independent research funded by the National Institute for Health and Care Research (NIHR) under its Programme Grants for Applied Research Programme (Grant Reference Number RP-PG-0618-20002; https://www.nihr.ac.uk/explore-nihr/funding-programmes/programme-grants-for-applied-research.htm), awarded to authors Lucy Yardley, Kay Wang and Mike Thomas. STC was supported by the National Institute for Health Research (NIHR) Health Protection Research Unit (HPRU) in Healthcare Associated Infections and Antimicrobial Resistance, a partnership between the UK Health Security Agency (UKHSA) and the University of Oxford (NIHR200915). The funder had no role in the design of the study; in the collection, analyses, or interpretation of data; in the writing of the manuscript, or in the decision to publish.

**Competing interests:** I have read the journal's policy and the authors of this manuscript have the following competing interests: MHVV is currently employed by Evinova (a separate health-tech business within AstraZeneca).

**Abbreviations:** BTS, British Thoracic Society; CDSS, Clinical Decision Support System; CFIR, Consolidated Framework for Implementation Research; COREQ, Consolidated criteria for reporting qualitative research; ERIC, Expert Recommendations for Implementing Change; FeNO, Fractional exhaled Nitric Oxide; NICE, National Institute for Health and Care Excellence; SIGN, : Scottish Intercollegiate Guidelines Network; StaRI, Standards for Reporting Implementation Studies.

especially when you can't see how they were produced; and Potential for the CDSS to increase patients' trust and adherence to their treatment. Clinicians perceived the CDSS could help them prioritise management options and consider broader factors relating to patients' asthma symptoms, but it was important to be able to override the recommendation. Lack of transparency over how recommendations were generated and concern about appropriateness of recommendations for specific patients led to uncertainty about adhering to the CDSS. Clinically tailored recommendations were perceived to help reassure patients and/or to support their adherence to asthma management.

## Conclusions

Even small changes to the content of CDSS recommendations, such as explaining how recommendations were generated and showing they are consistent with guidance, may help to overcome barriers to acceptability and perceived appropriateness for clinicians. Focusing on implementation during the development of CDSS interventions is worthwhile to help reduce the evidence-practice gap.

## Introduction

Clinical decision support systems (CDSSs) aim to 'improve healthcare delivery by enhancing medical decisions with targeted clinical knowledge, patient information, and other health information' [1,2]. They can promote adherence to guidelines, improve patient outcomes, and support clinical decision-making [1,3–5]. CDSSs have been shown to be effective [6], but implementation in terms of fidelity of delivery (also known as adherence), acceptability, and perceived appropriateness is often sub-optimal, resulting in recommendations being over-ridden or poor uptake of the CDSS in practice [3,7]. This suggests that the way in which these interventions are designed for and introduced to clinical practice could be improved.

Implementation strategies are "methods or techniques used to enhance the adoption, implementation, and sustainability of a clinical program or practice" [8]. The selection of relevant strategies for a specific intervention and context can be informed by theory, evidence around the determinants of implementation (defined as "modifiable factors that prevent or enable the adoption and implementation of evidence-based interventions" [9]), and stakeholder engagement [10]. A framework for implementation of CDSSs has been developed which provides an overview of six positions that clinicians may adopt, with tailored implementation strategies for each [11]. These positions range along a spectrum from perceiving low control over the CDSS, where it is seen as an interference or restriction on clinical decision making, to high control where it is seen as a helpful tool to complement clinical practice. The implementation strategies generally focus on what can be done within the organisation to facilitate the adoption of CDSSs, such as securing management commitment, integrating with existing processes, and involving clinicians in selecting sources of evidence on which CDSS recommendations are based.

This focus on changing the set-up and the setting *around* the intervention to promote implementation is reflected outside the CDSS domain, for example in the 73 implementation strategies listed in the ERIC taxonomy (Expert Recommendations for Implementing Change), which describe strategies that can be used at the organisation or user level to improve implementation of an intervention [12].

However, we believe that optimal design of the intervention itself can also impact on implementation outcomes. Indeed 'intervention' is included as one of five domains associated with implementation in the Consolidated Framework for Implementation Research (CFIR), along with inner and outer setting, process, and individual characteristics (24, 25). Recent recommendations have called for implementation researchers to focus on ways to optimise implementation at the level of the healthcare intervention, as well as at the clinician and organisational level (29). Explicitly identifying and describing implementation strategies during early phases of intervention research, such as development or feasibility studies, provides the opportunity to consider how to optimise implementation while the intervention is still under development [13], thus helping to address the evidence-to-practice gap [13,14]. Therefore, this study sought to identify ways that CDSS interventions can be optimised to facilitate their implementation.

The CDSS in this study was an online system for use during asthma reviews in primary care. Its aim was to reduce asthma exacerbations through the incorporation of Fractional exhaled Nitric Oxide (FeNO) test results to clinical decision making about asthma management [15]. FeNO tests are used to assess airway inflammation during asthma diagnosis, and can provide a more accurate indication of exacerbation risk than relying on patient-reported symptoms and lung function assessments alone [16,17]. The National Institute for Health and Care Excellence (NICE) have called for evidence to support the use of FeNO testing in improving asthma management (3). The CDSS was based on the latest evidence, drew on theory to change clinicians' beliefs about the benefit of FeNO testing, and was developed using the Person-Based Approach [18] with regular consultation with target users and stakeholders.

This study sought to explore how clinicians perceive the use of a CDSS intervention in asthma consultations, and how CDSS interventions can be optimised to support their implementation into practice.

## Methods

### Design

Two qualitative studies were conducted in primary care in the UK; a think-aloud study which took place in iterative stages from 01 July 2020 to 30 April 2021 to inform intervention development, and a process study from 01 August to 08 December 2021 which was nested within a feasibility study, see Fig 1. This design enabled understanding of clinicians' perceptions of the CDSS in two contexts; real-time in a hypothetical context during intervention development, and retrospectively when recalling use during asthma reviews. Optimisations were made to the CDSS intervention throughout the think-aloud and feasibility phases.

### Ethics approval and consent to participate

Ethical approval was granted by National Health Service Research Ethics Committees for both studies (South Central–Berkshire B Research Ethics Committee, 20/SC/0235, and Northwest —Greater Manchester East Research Ethics Committee, 21/NW/0078, respectively).

This paper reports the findings of these studies following the consolidated criteria for reporting qualitative research (COREQ) checklist [19] (S1 File) and the Standards for Reporting Implementation Studies (StaRI) statement [20] (S2 File).

### Intervention (the CDSS)

The CDSS was designed for use by clinicians during asthma reviews. It included web pages for clinicians to input information about their patient: FeNO test result (an measure of steroid-

**Fig 1. Study design and timelines.**

responsive airway inflammation); Asthma Control Test score (self-reported symptoms); and presence or absence of asthma exacerbations in the last 12 months (exacerbation risk). The CDSS then asked a series of tailored questions according to the data inputted, such as whether the patient was adherent to their treatment, and whether they were already prescribed certain treatments, before producing a tailored recommendation for the patient's care.

The CDSS recommendations were based on an algorithm developed by leading clinicians with expertise in FeNO-guided asthma management (KW, MT and others) via consensus meetings. They determined how the available evidence should be applied to interpreting FeNO tests for asthma monitoring, taking account of British Thoracic Society (BTS) and NICE guidelines [21,22].

Fig 2 shows a screenshot of the CDSS.

Before using the CDSS, clinicians completed an online training session about FeNO testing and how to use the CDSS during asthma reviews. The training session and the wording for the CDSS's tailored recommendations were co-developed with in-depth involvement from patients with asthma and clinicians, using the person-based approach [23].

Table 1 shows examples of possible recommendations received by clinicians from the CDSS.

## Context

GP practices in the UK invite patients with asthma to attend an annual review, and this is where the CDSS was designed to be used. All general practices in the feasibility study were provided with a FeNO analyser. At the time of this study, FeNO testing was not implemented as standard care for asthma management in Primary Care in the UK, but some clinicians have experience of using it to help diagnose asthma [24].

At the time of the think-aloud and feasibility studies (July 2020-December 2021), Covid-19 was still a significant concern which likely impacted how practices managed and conducted asthma reviews. Due to limited capacity and to minimise unnecessary face to face contact, practices tended to conduct asthma reviews remotely where possible and limit face to face asthma reviews, but patients were able to attend the practice in person to use the FeNO analyser. If no FeNO test result was available, the CDSS could not be used as it required a FeNO result to be inputted in order to generate a recommendation.

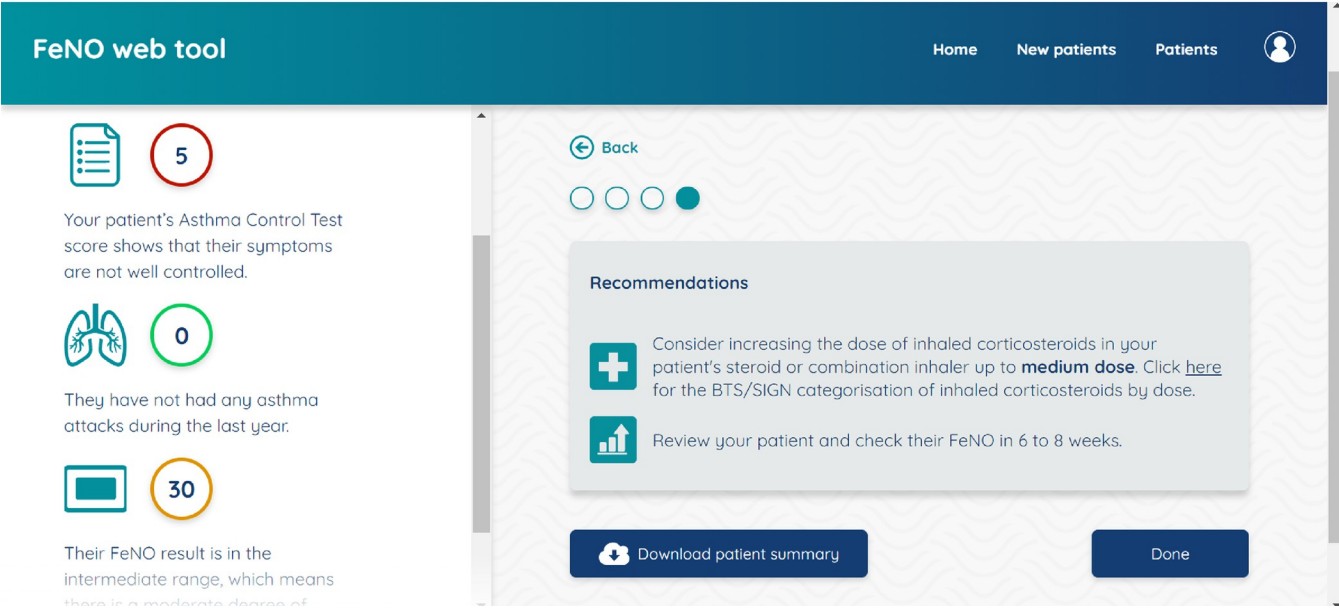

**Fig 2. Screenshot of the CDSS used in this study.**

## Recruitment

Clinicians managing patients with asthma in primary care were eligible to take part in the think-aloud and feasibility study.

**Table 1. Examples of clinically tailored CDSS recommendations.**

| Patient information | | | Recommendations developed by stakeholders as a decision tree |
|---|---|---|---|
| FeNO test result (low indicates absence of airway inflammation, high indicates airway inflammation) | Patient has experienced at least one exacerbation in the last year | Patient-reported asthma symptoms | |
| Low | No | Well-controlled | • If your patient's symptoms have been well controlled for 3 months or longer, consider stopping their leukotriene receptor antagonist.<br>• Review your patient and check their FeNO in 6 to 8 weeks.<br>• Advise them to seek medical advice sooner if they feel their asthma is getting worse. |
| Low | No | Poorly-controlled | • Ensure that any other factors which may be making your patient's asthma symptoms worse are adequately addressed (or refer to a colleague who can do this).<br>• Other strategies may be useful and should be tailored to your patient (Link)<br>• Once these factors have been addressed adequately, review the patient again and check their FeNO. |
| Intermediate | No | Well-controlled | • Consider starting your patient on low dose inhaled corticosteroids<br>• Review your patient and check their FeNO in 6 to 8 weeks |
| High | Yes | Well-controlled | • Consider increasing the dose of inhaled corticosteroids in your patient's steroid or combination inhaler back to the dose they were taking before.<br>• Review your patient and check their FeNO in 6 to 8 weeks |

### Think-aloud interviews

Recruitment to the think-aloud interviews was supported by the Thames Valley and South Midlands Clinical Research Networks. A range of clinicians were invited by email through mailouts to practices within the networks, including GPs, nurses, and pharmacists. In line with the person-based approach to developing digital behavioural interventions [25], recruitment took place in iterative cycles of 4–5 interviews with optimisations made to the CDSS recommendations and online training after each batch of interviews. This allows the study team to adapt and optimise interventions and explore the acceptability of such optimisations. Recruitment stopped once no new barriers to using the intervention were raised, in line with the person-based approach [25].

### Feasibility study process interviews

Recruitment to the feasibility study was supported by the Wessex Clinical Research Network. Where possible practices with no experience of using FeNO testing in asthma reviews were purposively sampled to explore how the intervention was implemented by novice users.

The six participating practices nominated a member of staff to use the CDSS during routine asthma reviews, usually an asthma or respiratory nurse, who was then invited by email to a process interview. All eligible staff involved in delivering the intervention in the feasibility study agreed to an interview.

### Procedure

In both studies, participant information sheets were sent by email, and informed consent recorded verbally prior to the interview. All interviews took place either by telephone or MS Teams. The interviews were conducted by MS (PhD), MV (PhD) and KM (PhD), female qualitative researchers at the Universities of Oxford and Southampton with experience conducting semi-structured interviews. Participants knew that the interviewers were researchers involved in the study, but were encouraged to share any negative views about the intervention. GP practices were reimbursed for clinicians' time. Interviews were audio-recorded and transcribed verbatim, but transcripts were not returned to participants to check.

### Think-aloud interviews

Participants were asked to think-aloud as they viewed the online training session, and/or used the CDSS interface to receive recommendations with dummy patient data. Open questions explored their experiences with a focus on understanding what was liked or disliked, and any perceived barriers or facilitators to implementing the intervention in practice. These interviews provided a rich complement to the process interviews by giving real-time, detailed perceptions of the CDSS content and concerns about implementing it in practice.

### Feasibility study process interviews

Semi-structured interviews explored participants' experiences of implementing the intervention during the feasibility study (see S3 File for interview schedule). Each clinician had conducted four or five asthma reviews using the CDSS at the time of the interview.

### Analysis

Reflexive thematic analysis [26] was used to analyse the process and think-aloud interview transcripts. This approach encourages themes to be built inductively from the data, influenced by the researchers' own interpretations, facilitating an in-depth understanding of perceptions

about the CDSS. The process and think-aloud interviews were analysed together as the data were found to contain similar themes, but the context in which the data were collected was considered when writing up the analysis and quotes are shown as think-aloud (T) or process (P) interviews to facilitate interpretation. KM led the thematic analysis, with feedback from STC, MS and BA to refine the development of themes.

All transcripts were read thoroughly to become familiar with the data, and codes were developed to identify meaning relevant to the research question. NVivo was used to capture the coding [27]. The researcher also kept a log during this process to record possible interpretations of the data. Themes were developed by interpreting shared meaning across codes in an iterative process, with ongoing revisions to the description of themes. Participants did not provide feedback on the analysis.

## Results

Sixteen think-aloud interviews in four batches, and seven qualitative process interviews were conducted. All clinicians in the feasibility study agreed to take part in an interview. We do not know how many clinicians were approached by the Clinical Research Network for think-aloud interviews but declined. During the think-aloud interviews, eight clinicians viewed just the online training module with patient scenarios and CDSS recommendations, six only inputted dummy data to the CDSS to view recommendations, and two did both. The average length of interviews was 46 minutes (range 21–86 minutes).

Participants in the thinkaloud interviews included 5 asthma nurses, 4 respiratory nurse specialists or nurse prescribers, 4 GPs and 3 pharmacists. Participants in the feasibility study included 3 asthma nurses, 2 respiratory nurse specialists or nurse prescribers, and 2 research nurses. Nineteen of the 23 participants had not used FeNO testing regularly prior to the study, whilst four had used it regularly. The details of the changes made to the CDSS between batches of thinkaloud interviews are reported elsewhere [23].

Three themes were developed: Appreciating the recommendations of the CDSS, whilst wanting to retain control; Doubt about appropriateness of CDSS recommendations, especially when you can't see how they were produced; and Potential for the CDSS to increase patients' trust and adherence to their treatment.

1. **Appreciating the recommendations of the CDSS, whilst wanting to retain control.** Some clinicians in the think-aloud interviews liked the idea of using a CDSS to help prioritise which management option to try, and facilitate decision-making in asthma reviews.

   *"Because I think sometimes it would have been a toss-up between reducing the dose of the inhaler first or stopping the additional leukotriene [sic], and who knows which one I would have picked?" (T15, pharmacist)*

   *"I think that's helpful guidance, it could even be like if it knew what your patient was on and the dose, if it knew what your local formulary was at CCG [clinical commissioning group] guidelines, that would be amazing if it then said exactly what you need to prescribe." (T16, GP)*

Clinicians felt that the tailored questions and recommendations from the CDSS could encourage them to consider additional factors when evaluating patients' asthma. For example, a nurse prescriber in a think-aloud interview felt that the CDSS would remind them to consider what else could be contributing to their patient's symptoms, rather than simply increasing asthma medication.

"*You've got somebody with what appears to be uncontrolled asthma symptoms, so rather than just ploughing on, upping the asthma treatment, this idea to really stop and think about is it the asthma or is it something else that's causing it? (T2, nurse prescriber)*

A GP also noted that *"it is good and it's making you think about other things, it's making you think about other causes of cough" (T3, GP).*

While the CDSS is intended to facilitate conversation with the patient with the ultimate decision about asthma management remaining with the clinician and patient, some participants in the thinkaloud interviews seemed to regard it more as a direction to be followed. A lead practice nurse in a respiratory hub felt that the CDSS would change their usual way of working and was unsure about being "told what to do".

"*It's interesting being told what to do by a tool, I think that's something that would be very new to us because I think that's kind of our job" (T13, Lead practice nurse and respiratory hub nurse)*

A GP considered that adherence to the recommendations might be higher amongst nurses than GPs.

"*Nurses are slightly better than doctors at following processes, we go off-piste too much and nurses would probably do very well with this and you can follow it, you can't go too far wrong with it." (T16, GP)*

However, when the CDSS was put into practice in the feasibility study, it appeared that clinicians did regard recommendations merely as possible actions which they could decide whether to follow, although some wanted to record their rationale if they chose to do something differently. This appeared to be more about wanting to justify their decision to the CDSS, or whoever monitors it, rather than for their own use or patient benefit.

"*It might be good to have some way of making note of that when you... They're obviously on our notes for the patient and we know why, but it might be an idea to do a little training session on what to do if you disagree with the recommendations, because I wasn't really quite sure how to go about that" (P1.1, research nurse)*

Other clinicians in the feasibility study felt that the CDSS did not add anything for them because they already had sufficient experience and knowledge to make decisions without it. These clinicians did not feel the need to explain when they had not followed its recommendations.

"*I: Were there any cases where the [CDSS] influenced your clinical decision making do you think?*

*R: No. [laugh] I think I went against one of them because you know, it just...I think if somebody is or has little experience, I won't say inexperienced, so somebody who is starting their journey they probably would find it a lot more useful... You know I'm very happy to admit I am an old nurse, I've been doing this virtually for 20 years. [laugh]" (P5, asthma nurse).*

Another clinician in the feasibility study agreed that the CDSS would be more useful for clinicians with less experience of asthma management, highlighting that the suggestion to consider non-pharmacological management options would be particularly beneficial.

**2. Doubt about appropriateness of CDSS recommendations, especially when you can't see how they were produced.**   Lack of transparency within the recommendations about how they were produced and which factors they took into account could lead clinicians to doubt their appropriateness to action. Recommendations were originally designed to be short and focused only on the action to take rather than how the algorithm had arrived at that conclusion, e.g. 'Ensure any other factors which may be making your patient's asthma worse are adequately addressed, or refer to a colleague who can do this'. However, seeing this particular recommendation in practice, one clinician concluded that it was just generic advice which was not that useful, and had not taken the patient's FeNO test result into account (even though it had):

"*We know all that as an asthma nurse–you know that's the basics….. I tended to then undo that, and just concentrate on the FeNO*" *(P4, asthma nurse).*

Another clinician overrode a recommendation to increase their patient's medication because the patient had reported having well-controlled symptoms. This clinician decided that the FeNO test must be wrong:

"*We know FeNO tests aren't 100% accurate, we know that you can get false positives and that there are other factors that can influence the FeNO result. I think having the FeNO result is obviously helpful in some cases, but may also be a little bit of a red herring*" *(P6, respiratory lead)*

These findings suggested that the brevity of the recommendations was a hindrance to their implementation, due to the lack of context and rationale.

However, other times clinicians perceived the recommended action to be insufficient or unwise, not so much due to the wording but rather because it was generated by an automated algorithm that could not understand the individual patient as well as they could. For example, this clinician described how the CDSS recommended a behavioural intervention but they felt that medication was needed for this patient:

"*It wasn't one that the computer had suggested but it's one that I thought we needed to do*" *(P1.2, asthma nurse).*

Other clinicians felt that the CDSS did not consider contextual factors, such as the pandemic, time of year, changes in exercise, or patient anxiety. The need for a holistic approach was perceived as particularly important when the CDSS recommended stepping down medication, for safety reasons.

"*Because I know the patients and how [long] it's taken us to get to the level that they are and in view of the climate that we're in with the pandemic and everything and us not knowing quite how the flu season is going to be this year, I've continued them on the treatment they're currently on with a view to maybe stepping down next spring*" *(P1.2, asthma nurse).*

A GP discussed a more general reticence amongst patients and staff to reduce medication once asthma is well controlled, due to low perceived need and benefit, which could reduce acceptability of this recommendation from the CDSS in practice.

"*Once they've been in once and they're stable and they're happy, quite a lot of the time it's more important that they just stable rather than be cut down, 'cause what are they saving?*

*They may be saving a tablet or a puff or two but they'd rather just make sure everything's fine and they're not rocking the boat the next six months. . ... you hear that from the nurses sometimes as well. They'll just be like, well look I'd rather not unless it's someone's who's very, very anti-medication." (T1, GP).*

**3. Potential for the CDSS to increase patients' trust and adherence to their treatment.**
Some clinicians felt that the CDSS recommendations could enhance patients' understanding of their asthma management and help engage them in the recommended approach for managing their asthma.

*"I would happily use this at work because I think it would make a difference, particularly in terms of explanation for the patients and helping them to come onboard with any changes that we might feel we need to make". (T5, asthma nurse)*

One clinician in the feasibility study described how seeing a recommendation which was consistent with the existing treatment plan could be reassuring for them and their patients.

*"One of my patients who is already under respiratory. . . when we put all the information in, it came out, 'Have you considered referring to Respiratory?!' and that was good to see because then, she said, 'Well, yeah, the machine is picking up the information that we're putting in,' so that was a good thing–it just clarified what she was already under" (P2, research nurse)*

Clinicians wanted the option to add their own notes to the CDSS's recommendations to optimise this potential for enhancing patient engagement.

*"And it might be a helpful tool actually to be able to add your own notes to then print off to give to the patient. That could form part of their management plan if you put on there" (P3, respiratory nurse specialist)*

## Discussion

The findings of this study show how understanding clinicians' experiences and perceptions of a CDSS can identify barriers to implementation. Specifically, concerns about retaining control to disregard recommendations, doubts about appropriateness due to lack of transparency about how recommendations were generated, and concerns about the recommendation not taking account of the wider clinical context were potential barriers. Based on these findings, we identified several strategies for optimising the implementation of CDSS interventions in healthcare settings through adapting the design and content of the CDSS itself rather than focusing only on the processes around the intervention. These theoretically-informed optimisations are discussed further below.

Specifically in this study, we implemented three implementation strategies to promote acceptability and perceived appropriateness of the CDSS intervention, based on the think-aloud and feasibility study process interviews. Firstly, we wanted to ensure it was clear to clinicians that they could override CDSS recommendations if they perceived them to be inappropriate. We added an open text- box to the CDSS for clinicians to record their decision-making, with an explanation that if a clinician and patient decided the recommendation was not right, they can record the reason why, in order to reinforce that the recommendation can be adapted.

Preserving clinician autonomy by ensuring CDSSs are not seen as prescriptive has been recognised as a priority for effective implementation [1,28–30]. The concept of 'negotiation of control' has been used to explain this process, whereby if clinicians perceive that a CDSS is

dictating a course of action, this can impinge on their professional autonomy and identity, reducing the likelihood of fidelity [11]. Furthermore, the concept of 100% adherence to a CDSS is usually neither realistic nor desirable, as this would conflict with the need to adapt standardised evidence-based recommendations to enable provision of holistic patient-centred care [31].

The design of the CDSS can facilitate the desired flexibility for clinicians, giving them 'permission' to make their own decision. For example, allowing clinicians to record the action agreed with the patient could help frame the CDSS as an advisory tool which informs you what the guidelines or evidence would recommend but with the option to adjust as necessary. While this may increase perceived acceptability, it could risk compromising adherence to the CDSS recommendations, also known as fidelity [32], as emphasising choice may mean CDSSs fail to promote adherence to guidelines in the very circumstances they are most needed, such as stepping down medication. Therefore, it is important to balance adaptability of recommendations while still promoting adherence when appropriate. Specific guidelines around when it may be acceptable to override a recommendation might help to facilitate appropriate adherence [3]. A further challenge for implementation research is defining a reasonable target for adherence to recommendations from CDSSs within a particular setting, in order to know whether a CDSS is being successfully implemented with appropriate adjustments, or whether there are issues with adherence.

Secondly, we worked with our stakeholders to add information to each recommendation about how it was generated, reassurance that it was in line with evidence, and to explicitly acknowledge and address perceived discrepancies, e.g. "Although your patient seems to have well-controlled symptoms, their FeNO result shows there may be some inflammation in their upper airways. Therefore. . ..". In addition, a full table of the potential CDSS recommendations a clinician might see was added to the training session, showing the circumstances in which each recommendation would arise to demonstrate all the factors which are taken into account.

The need for transparency about how CDSSs generate recommendations is important for clinicians, who need to be able to take responsibility for their clinical decisions [33]. This is consistent with research showing that clinicians need to understand the evidence behind a CDSS in order to trust it [11,34]. Therefore, providing a clear rationale alongside the action being recommended is essential to promote acceptability, fidelity and perceived appropriateness. This rationale might include fit with evidence and guidelines, and details about the information that was used to produce the clinically tailored recommendation. In this case, the additional information was incorporated within the main recommendation, whilst another CDSS included these details as optional additional information that can be viewed by those who want to read more [35], although clinicians in busy settings may not have time to engage with this. A challenge relating to this implementation strategy will be ensuring that the CDSS can be updated based on newly emerging evidence and guidelines [11].

Finally, during training, and within the CDSS recommendations we explained how use of the CDSS enhances patient outcomes, and added content to the interface to facilitate discussion with patients about the recommendations during the consultation. The tendency for more experienced clinicians to perceive a low need for CDSSs has been reported previously [11,29,35,36]. While this low perceived need might be addressed by first-hand experience of the benefits the CDSS can offer [11], this study found it remained a barrier even when clinicians had used the CDSS in practice. The issue appeared to be that the CDSS was not perceived to add anything to their own clinical judgment, defined by the CFIR as 'relative advantage'. The CFIR suggests relative advantage can be promoted by visibly demonstrating the benefits of the intervention [37], and while the online training in this study included details about how the CDSS could improve patient outcomes in different clinical scenarios, the credibility and

impact of this message might be increased through top-down driven change to show that senior leaders or managers endorse the intervention [11]. Other strategies to visibly demonstrate relative advantage may include showing how the CDSS has helped improve patient outcomes in other sites, or encouraging clinicians to discuss recommendations with their patients, to show how this can positively impact on patients' reassurance, motivation, or receptiveness to certain management recommendations [38]. Indeed, the potential for CDSSs to help patients better understand their condition has been recognised by clinicians as a benefit to their use in practice [35].

Fig 3 shows a logic model representing how the optimisations made to the CDSS following these interviews acted as implementation strategies, and possible contextual factors that could impact on how these strategies operate.

S4 File provides more detail about how these strategies are theorised to work, and how they map on to implementation outcomes and taxonomies.

In terms of implementation research, this study suggests it is valuable to explore implementation during the early stages of intervention development and evaluation, in order to understand how implementation strategies could be built into the intervention itself, potentially reducing the risk of an evidence-practice gap [39]. Indeed, even small changes to the CDSS recommendations may be quite powerful for addressing concerns about acceptability and perceived appropriateness for clinicians. This study supports the recommendation that CDSSs need to be designed with consideration of the complex process and setting in which they will

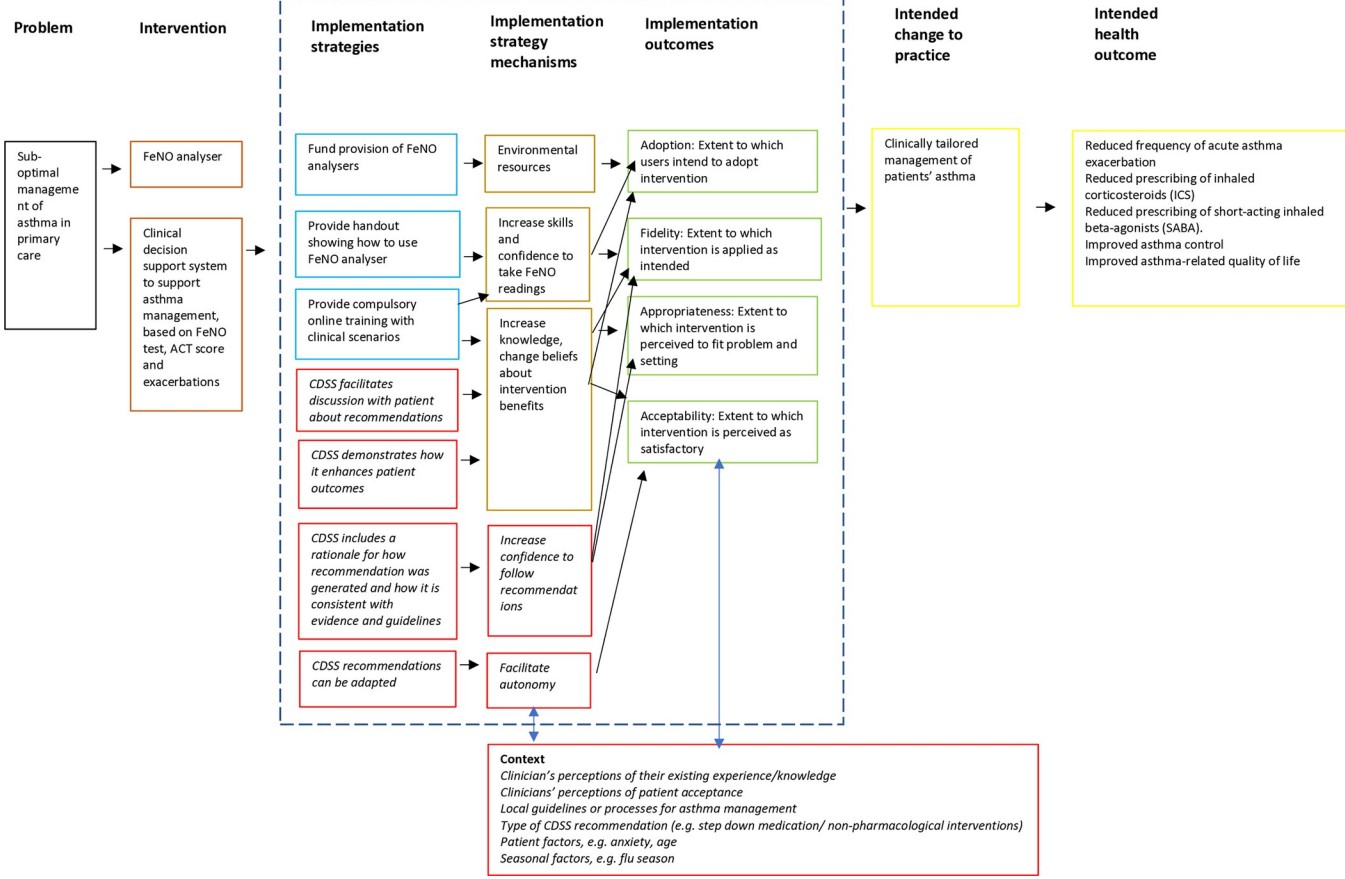

**Fig 3. Logic model showing CDSS optimisations (*italics*) to promote implementation and contextual factors influencing implementation.**

be used, drawing on implementation theories to best understand how to optimise this process [11]. The transferability of the implementation strategies proposed by this study for other CDSS requires further exploration, but we propose there may be value in identifying common implementation strategies for interventions which share characteristics or purpose.

A strength of this research is that the feasibility study enabled perceptions of real-life implementation of a CDSS to be explored, rather than just hypothetical factors. Also, although the recommendations for optimising CDSSs to promote implementation were developed within the context of asthma management, the implementation determinants resonated with broader CDSS research suggesting that the optimisations could have wider relevance. However, the sample size was relatively small with only six GP Practices implementing the CDSS, and most clinicians were asthma nurses which restricted our ability to explore contextual variations between sites and roles that might influence implementation outcomes. Furthermore, the process interviews relied on retrospective considerations about how the CDSS was used, whereas real-time observations of patient-clinician interactions using the CDSS could offer valuable insights into factors influencing fidelity, acceptability and perceived appropriateness which may not be recalled or even consciously noticed by clinicians. While the think-aloud interview participants did not use the intervention in a real-life setting, these real-time reflections about the CDSS rationale and content of the recommendations were very valuable in understanding possible barriers to implementation.

## Conclusions

This paper recommends that CDSSs could promote acceptability, perceived appropriateness and fidelity by enabling alternative actions to be recorded where clinicians decide to follow a different management plan, showing clinicians how recommendations for patient care were generated, including reminders to show recommendations are consistent with guidance, and encouraging clinicians to discuss CDSS recommendations with patients. Considering implementation strategies early on during intervention development and evaluation can enable the optimisation of interventions to incorporate strategies which promote successful implementation.

## Supporting information

**S1 File. Consolidated criteria for reporting qualitative research (COREQ) checklist.** (DOCX)

**S2 File. Standards for Reporting Implementation Studies (StaRI) statement.** (DOCX)

**S3 File. Feasibility study qualitative interview schedule.** (DOCX)

**S4 File. Implementation strategies identified from this analysis.** (DOCX)

## Author Contributions

**Conceptualization:** Lucy Yardley, Mike Thomas, Kay Wang, Ben Ainsworth, Sarah Tonkin-Crine.

**Data curation:** Marta Santillo, Michelle Helena Van Velthoven.

**Formal analysis:** Kate Morton, Sarah Tonkin-Crine.

**Methodology:** Kate Morton, Lucy Yardley, Kay Wang, Ben Ainsworth, Sarah Tonkin-Crine.

**Project administration:** Kate Morton, Marta Santillo, Michelle Helena Van Velthoven, Kay Wang, Sarah Tonkin-Crine.

**Writing – original draft:** Kate Morton.

**Writing – review & editing:** Kate Morton, Marta Santillo, Michelle Helena Van Velthoven, Lucy Yardley, Mike Thomas, Kay Wang, Ben Ainsworth, Sarah Tonkin-Crine.

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
