## [Decision Letter · Decision Letter 0]

13 Aug 2024

PONE-D-24-07315Promoting the implementation of clinical decision support systems in primary care:

A qualitative exploration of implementing a Fractional Exhaled Nitric Oxide (FeNO)-guided decision support system in asthma consultationsPLOS ONE

Dear Dr. Morton,

Thank you for submitting your manuscript to PLOS ONE. After careful consideration, we feel that it has merit but does not fully meet PLOS ONE’s publication criteria as it currently stands. Therefore, we invite you to submit a revised version of the manuscript that addresses the points raised during the review process. Reviewer number 2 has made some suggestions.  There are mostly requests for clarification, and I agree that these changes will result in an improved manuscript.

We look forward to receiving your revised manuscript.

Kind regards,

Dolly Baliunas

Academic Editor

PLOS ONE

Reviewers' comments:

Reviewer's Responses to Questions

**Comments to the Author**

1. Is the manuscript technically sound, and do the data support the conclusions?

Reviewer #1: Yes

Reviewer #2: Yes

2. Has the statistical analysis been performed appropriately and rigorously? 

Reviewer #1: N/A

Reviewer #2: N/A

3. Have the authors made all data underlying the findings in their manuscript fully available?

Reviewer #1: No

Reviewer #2: No

4. Is the manuscript presented in an intelligible fashion and written in standard English?

Reviewer #1: Yes

Reviewer #2: Yes

5. Review Comments to the Author

Reviewer #1: Excellent paper - clear, comprehensive, informative, well written. The topic is of great interest. The methodology can be used by other clinicians testing implementation strategies of CDSS in other clinical context.

Reviewer #2: Interesting paper regarding the development and feasibility testing of a CDSS for FeNO testing.

Please find my comments in the attached document. In general, there's a need to expand re how the CDSS works, include more diverse quotes especially GPs that seem not represented and place the study in the wider context of similar Think Aloud studies recently done.

6. PLOS authors have the option to publish the peer review history of their article (what does this mean?). If published, this will include your full peer review and any attached files.

Reviewer #1: No

Reviewer #2: No

---

## [Author Response · Author response to Decision Letter 0]

20 Nov 2024

Dear Reviewers,

Thank you very much for reviewing this paper, we really appreciate your additional suggestions for improvements and have implemented them below. The page and line numbers refer to the location of the change in the tracked copy. 

Many thanks

Kate Morton

Corresponding author

Were these rule based recommendations e.g. based on a decision tree, or generated via other means e.g. Generative AI? 

Table 1, page 8, methods T

he recommendations were based on a decision tree. This is referred to on page 7:

“The CDSS recommendations were based on an algorithm developed by leading clinicians with expertise in FeNO-guided asthma management (KW, MT and others) via consensus meetings”.

I’ve also made this clearer in Table 1 by renaming the column that was called ‘recommendations’ as ‘recommendations developed by stakeholders as a decision tree’ (page 8)

What happens to the CDSS if there's no FeNO results available? 

Page 9, methods 

Clarified that ‘If no FeNO test result was available, the CDSS could not be used as it required a FeNO result to be inputted in order to generate a recommendation.’ (page 9)

Why 4-5 interviews the cut-off for one cycle? 

Page 9-10, methods 

This batched approach is in line with the person-based approach to developing digital behavioural interventions. It allows the study team to adapt and optimise interventions and explore the acceptability of such optimisations. A citation has been added to clarify this (page 9-10)

How many Think Aloud cycles were done and what was the change across the cycles? 

Page 12, methods 

There were 4 cycles of thinkaloud interviews, and this has been added to the paper (page 12).

Changes made to the CDSS between batches of thinkaloud interviews are reported elsewhere, and this has been clarified (page 12)

Clarify this breakdown. Does this mean 8 clinicians did not had a direct interaction with the CDSS screens but only the online training module? 

Page 12, results 

This has now been clarified: 

“eight clinicians viewed just the online training module with patient scenarios and CDSS recommendations, six only inputted dummy data to the CDSS to view recommendations, and two did both”. (page 12)

Suggest a breakdown across the 2 parts - Think Aloud and Process evaluation. What were their experiences in managing asthma, gender distribution, practice size, volume of asthma patients seen each month? 

Page 12, results 

We have now broken this down between the thinkaloud and feasibility study for the data we have, but unfortunately we did not collect data regarding experience in managing asthma, gender, practice size, or volume of asthma patients seen each month. (page 12)

No quotes from GPs seem to have been included, please ensure representation from all the groups is present and negative/dissenting opinions were also included 

Page 12, results 

Thank you for highlighting this. We have now included some more quotes from GPs to ensure their views are represented and discussed (pages 12-18)

How does these results fit within the wider CDSS implementation literature? Some recent relevant papers:

https://pubmed.ncbi.nlm.nih.gov/33965933/

https://pubmed.ncbi.nlm.nih.gov/35673136/

https://doi.org/10.3399/BJGP.2022.0608

https://pubmed.ncbi.nlm.nih.gov/36809791/

As well as when considering common models of technology adoption e.g. UTAUT and TAM. 

Page 17, discussion 

Thank you for the additional references, which have been read and added in to enhance the discussion and links with current literature (pages 18-22)

---

## [Editor Report · Decision Letter 1]

2 Jan 2025

Promoting the implementation of clinical decision support systems in primary care:

A qualitative exploration of implementing a Fractional Exhaled Nitric Oxide (FeNO)-guided decision support system in asthma consultations

PONE-D-24-07315R1

Dear Dr. Morton,

We’re pleased to inform you that your manuscript has been judged scientifically suitable for publication and will be formally accepted for publication once it meets all outstanding technical requirements.

Kind regards,

Marsa Gholamzadeh, PhD

Academic Editor

PLOS ONE
---

## [Editor Report · Acceptance letter]

14 Jan 2025

PONE-D-24-07315R1 

PLOS ONE

Dear Dr. Morton, 

I'm pleased to inform you that your manuscript has been deemed suitable for publication in PLOS ONE. Congratulations! Your manuscript is now being handed over to our production team.

Kind regards, 

on behalf of

Dr. Marsa Gholamzadeh 

Academic Editor

PLOS ONE